# OpenReview forum: "Patching LLM like Software: A Lightweight Method for improving existing policy in Large Language Models"
_ICLR.cc/2026/Conference — Submitted to ICLR 2026_

### Official Review · Reviewer_2NEg · 2025-10-27

**Soundness:** 2
**Presentation:** 4
**Contribution:** 3
**Rating:** 4
**Confidence:** 4

**Summary:**

This paper proposes a lightweight safety alignment method through prompt tuning. The authors assumed the scenario that a vendor cannot instantly update their models due to the high computational resource of full fine-tuning and parameter efficient tuning for every failure cases reported from customer. To resolve safety vulnerabilities (toxicity, bias, harmfulness) discovered major model releases, the authors prepend a learnable prefix with only 0.003% additional parameters to the model input. Specifically, they train a policy patch that learns the behavior of an improved model, which customers can then apply to the original model M to create M+. Training proceeds in two stages: SFT mimics M's token distribution, followed by DPO learning preferences between safe and unsafe responses. The method is validated across diverse model families including Llama, Aya, Mistral, and Gemma, demonstrating safety improvements.

**Strengths:**

1. The paper's Vendor-Customer scenario is practical and convincing, effectively reflecting real-world industry environments. Full-model fine-tuning (RLHF, DPO) is computationally expensive and inaccessible. Major model releases occur only once or twice a year, making it impossible to immediately address known safety issues. Learning soft prompt can be good approach for handling this scenario.

2. The authors addressed various major safety risk experiments, including toxicity, bias, harmfulness. The proposed method is validated with consistency across diverse model families (Llama-2/3, Aya-23, Mistral, Gemma2, Vicuna). Also, the several ablation studies show the impact of \beta value, patch length, and initialization methods.

3. Using multiple learned prefix together shows the significance of this approach. By solely combining multiple prompts from toxicity and bias, the proposed method can address both safety risk scenarios even though these prompts are individually trained.

4. As shown in Table 2, the Policy Patch (0.2M parameters, +2.5% inference overhead) achieves ~70% toxicity reduction with 195× fewer parameters compared to LoRA (40M parameters, +24% inference overhead). This presents a practical trade-off from a deployment perspective.

**Weaknesses:**

1. The paper's most significant limitation is its dependency on an improved model M'. The authors assumed that M' (already safety-improved model) must pre-exist. Then, the learnable prefixes are trained on data generated by M'. Thus, the paper fails to actually solve the "gap between major version updates" that it claims to address. Current work only addresses the unknown issues  that a company cannot release a updated model. This is not a work addressing the parameter efficiency as it first obtains M' and then train prefix. While Section 5 Limitations mentions "reliance on an improved reference model," it does not sufficiently emphasize that this undermines the method's foundation.

2. The current method consists of well-established techniques from existing works. Method = Prompt Tuning + SFT  + DPO. The core contribution is applying these to safety problems and reinterpreting as a "patch" framework. However, there are already many works that addresses the safety issues based on prompt tuning [1]

[1] DP-OPT: Make Large Language Model Your Privacy-Preserving Prompt Engineer. ICLR 2024.

3. While I like the experiment in Table1, the current experiment include limited composition with Only Toxicity + Bias combination tested. To further support the authors' claim, conflict potential between patches should be examined. e.g., does the toxicity patch already address or worsen bias? Hence, I suggest the authors to add the experiments with harmfulness included and all three risks combined.

Minor typo in Line 242

**Questions:**

1. I would like to know the details of LoRA adapter in Table 2. What data did you use for SFT, and do you perform DPO together?

2. In Figure 2, M+ increase PPL compared to M' with 4 value difference. However. in line 302, it is described as it maintains PPL close to the aligned model M'? How can I interpret the result as maintained?

**Details Of Ethics Concerns:**

While this paper addresses ethically sensitive topics (toxicity, bias, harmfulness in LLMs), it does not require an ethics review for the following reasons. First, the paper contributes to make LLMs safer, not create  safety issues. This work focuses on reducing toxicity, bias, and harmful content generation. Moreover, the paper includes "ETHICS STATEMENT" section stating that it uses publicly available datasets without personally identifiable or sensitive private data, human subjects involved.

---

> ### Author Response · Authors · 2025-11-23
> **Author response: dependence on M′, novelty, and composition**
>
> We thank the reviewer for their thoughtful and constructive feedback, and for highlighting several aspects they found positive: the practical and convincing vendor–customer scenario, the broad coverage of safety risks (toxicity, bias, harmfulness) across diverse model families (Llama-2/3, Aya-23, Mistral, Gemma2, Vicuna), the informative ablations on β, patch length, and initialization, the empirical value of composing multiple prefixes, and the strong deployment trade-off versus LoRA (Table 2). We fully agree that the key open questions are (i) how strongly our approach depends on an improved model M′, (ii) how it differs from prior prompt-tuning–based safety work, and (iii) how patches interact when composed (and how this relates to harmfulness). In what follows, we directly address these points and answer the reviewer’s specific questions about the LoRA setup and PPL interpretation.

---

> ### Author Response · Authors · 2025-11-23
>
> > **Concern:** Since the method assumes an improved model ( \mathcal{M}' ), does it really help with the “gap between major releases” and parameter efficiency, or does it just move the problem to training many improved variants?
>
> > **Answer (Brief):** We only require *some* sufficiently safe reference model, not a bespoke improved variant per backbone. In the revision we show that a single safer checkpoint can be reused as a cross-teacher for multiple heterogeneous students (Aya-23, Llama-2, Llama-3), and the only thing we train per backbone is a tiny 0.003% prefix. Parameter efficiency thus sits entirely in the patching layer, not in (re)training (\mathcal{M}').
>
> **Detail**
>
> We thank the reviewer for highlighting this point. Our method does require access to **some** safer reference model, but it does **not** require training a new improved variant $\mathcal{M}'$ for *each* backbone we want to patch. Two clarifications are important:
>
> 1. **A single safer checkpoint can be reused as the safety reference.**
>
>    In the revision (App. A.16), we explicitly show that the reference model used to generate safe responses does *not* need to match the architecture of the model being patched. Using the RTP–Challenging setup, we patch Aya-23, Llama-2, and Llama-3 while taking safe responses from different existing checkpoints (Aya-23, Llama-3, Llama-2), as summarized below:
>
>    | Patched Model | Reference Model (Safety Guide) | Max Tox. | Tox. Rate | PPL | Div. |
>    |---------------|--------------------------------|----------|-----------|-----|------|
>    | Aya-23 | Aya-23 | 0.081 | 0.017 | 13.0 ± 4.5 | 0.123 |
>    | | Llama-3 | 0.097 | 0.041 | 12.4 ± 4.1 | 0.124 |
>    | | Llama-2 | 0.086 | 0.033 | 12.8 ± 5.1 | 0.129 |
>    | Llama-2 | Aya-23 | 0.188 | 0.083 | 10.9 ± 3.2 | 0.058 |
>    | | Llama-3 | 0.197 | 0.050 | 10.9 ± 3.2 | 0.057 |
>    | | Llama-2 | 0.247 | 0.183 | 10.8 ± 3.2 | 0.078 |
>    | Llama-3 | Aya-23 | 0.266 | 0.183 | 14.4 ± 5.0 | 0.059 |
>    | | Llama-3 | 0.296 | 0.233 | 13.9 ± 5.1 | 0.055 |
>    | | Llama-2 | 0.256 | 0.200 | 14.5 ± 5.3 | 0.053 |
>
>    For example, **Llama-2 as the patched model** achieves substantially lower Toxic Rate when guided by Aya-23 or Llama-3 than when guided by its "own" improved variant (0.083 and 0.050 vs. 0.183), with almost unchanged PPL.
>
>    This shows that **any sufficiently safe existing checkpoint can act as the safety policy guide**, and a vendor does *not* need to first obtain (or train) a bespoke improved version of every deployed backbone before our method becomes useful.
>
> 2. **Where parameter efficiency comes from.**
>
>    Once such a safer checkpoint exists (e.g., a prior internal model, a stronger hosted model, or a public safety-tuned release), our approach **does not change that model at all**. The *only* parameters that are trained are the tiny prefixes for the deployed models—0.003% of the backbone, +2.5% inference overhead in our main toxicity risk experiments.
>
>    In other words, **parameter efficiency is exactly in the patching layer**:
>    - One reasonably safe model can serve as a *fixed* safety reference,
>    - Many different deployed backbones can be updated by learning very small prefixes against that fixed reference, instead of running a new parameter-efficient fine-tuning (or full RLHF pipeline) for each backbone and risk.
>
> We now clarify this deployment picture in the paper: the method is not meant to be an alternative to training *some* strong safe model, but rather a parameter- and compute-efficient way to turn *any existing safer checkpoint* into lightweight patches that can be quickly shipped to multiple models between major updates.

---

> ### Author Response · Authors · 2025-11-23
>
> > **Concern:** The method mainly combines known components (prompt/prefix tuning + SFT + DPO), so its novelty seems limited, and prior work like DP-OPT already uses prompt tuning for safety-related goals.
>
> > **Answer (Brief):** We agree the primitives are standard, but our contribution is different in **artifact, setting, and application**: we introduce and empirically validate *continuous safety policy patches* as deployable, composable vendor-side artifacts (0.003% params) for multi-risk safety control across many backbones and strong attacks—rather than general task prompts or privacy-preserving prompts as in DP-OPT.
>
> **Detail**
>
> We appreciate the reviewer's comment and agree that our building blocks (SFT, DPO, prefix tuning) are individually well-established. Our contribution is *not* a new optimization algorithm, but a different **artifact, setting, and threat model** from prior prompt-tuning work such as DP-OPT.
>
> Concretely, our work differs in three ways:
>
> 1. **Artifact and deployment role.**
>
>    DP-OPT learns a *discrete textual prompt* under a **privacy** objective, with the main contribution being differential privacy guarantees for releasing that prompt to an external API.
>
>    Our work instead introduces a **safety policy patch**: a tiny *continuous* prefix (0.003% params) that is:
>    - plugged into the vendor's own backbone at inference time,
>    - treated as a *versioned safety artifact* that can be shipped independently of a model release, and
>    - explicitly compared against LoRA adapters as an alternative safety mechanism (Table 2).
>
> 2. **Problem setting and risk model.**
>
>    Our focus is **safety control under risky/adversarial inputs**, not privacy of the prompt:
>    - Three distinct risks: toxicity, professional-context gender bias, and harmfulness/refusals.
>    - Evaluated across four model families (Llama-2/3, Aya-23, Gemma-2, Mistral, Vicuna).
>    - New results (revision) include **multi-risk patches and composition** (Sec. 4.3) and **robustness to strong jailbreak attacks** (PAIR, GCG, JBC; App. A.18).
>
>    To our knowledge, prior prompt-tuning work (including DP-OPT) does *not* position the learned prompt as a reusable, vendor-side *safety patch* evaluated across this range of risks and architectures.
>
> 3. **"Patch" behavior and composition.**
>
>    We empirically study the **modularity and composability** of these prefixes:
>    - Single multi-risk patch ($P_{\text{multi}}$) jointly addressing toxicity + bias.
>    - Composed prefixes ($[P_{\text{tox}}, P_{\text{bias}}]$) applied as a *single* 100-token patch, with analysis of order sensitivity and trade-offs (Sec. 4.3, Table 6).
>
>    This "stackable safety patch" behavior is what enables the vendor-style "fix between major releases" scenario, which is conceptually different from one-off task prompts or privacy-preserving prompt publication.
>
> We build on standard SFT/DPO signals, but the **novelty lies in evaluating, and validating continuous prefixes as deployable, composable *safety policy patches*** rather than general task prompts or DP-protected instructions.

---

> ### Author Response · Authors · 2025-11-23
>
> > **Concern:** Composition is only tested on toxicity + bias, and there is no explicit study of conflict between patches
>
> > **Answer (Brief):** In the revision we **explicitly analyze cross-effects** between toxicity and bias patches on Llama-2-7B: the toxicity patch slightly *improves* bias, the bias patch can over-regularize toxicity (low diversity), and a composed patch ([P_{\text{tox}}, P_{\text{bias}}]) **improves both risks simultaneously** (strongest toxicity + better bias) without catastrophic interference. We also add a **single multi-risk patch (P_{\text{multi}})** trained jointly on toxicity + bias, which improves both risks and attains the best GLD.
>
>
> We agree that understanding *interactions* between patches is important. In the revision we keep the toxicity–bias case as our main composition study, but we now explicitly analyze conflict / cross-effects between patches rather than only reporting per-risk scores.
>
>
> A subset of the results (Table 6 in Sec 4.3) is:
>
> | Model Config.              | Avg Max Tox ↓ | Tox Rate ↓   | Avg GAS ↓   | Avg GLD ↓   | Div (Tox) ↓ | Div (Bias) ↓ |
> |----------------------------|---------------|--------------|-------------|-------------|-------------|--------------|
> | No P                       | 0.7809        | 0.5520       | 0.3400      | 0.7012      | 0.0437      | 0.0020       |
> | P₍tox₎                     | 0.0619        | **0.0040**   | 0.3040      | 0.3622      | 0.0156      | 0.0079       |
> | P₍bias₎                    | 0.0527        | **0.0000**   | **0.0120**  | 0.4082      | 0.5748      | 0.1119       |
> | P₍multi₎ (tox + bias)      | 0.1109        | 0.0160       | 0.1240      | **0.2521**  | 0.1660      | 0.0756       |
> | P₍comp (tox first)₎        | **0.0282**    | **0.0000**   | 0.0200      | 0.3700      | 0.0539      | 0.0509       |
>
>
>
>
> - **Does the toxicity patch already address or worsen bias?**
>   $P_{\text{tox}}$ slightly **improves** bias vs. No $P$ (GAS 0.3040 vs. 0.3400; GLD 0.3622 vs. 0.7012), but it does *not* come close to the strongly debiased behavior of $P_{\text{bias}}$ (GAS 0.0120). It does **not** worsen bias; rather, it yields limited collateral improvement.
>
> - **Does the bias patch hurt toxicity?**
>   $P_{\text{bias}}$ achieves good toxicity scores numerically (Avg Max Tox 0.0527, Tox Rate 0.0), but at the cost of **very poor diversity** on toxicity prompts (trigram overlap 0.5748 vs. 0.0156 for $P_{\text{tox}}$), indicating repetitive, template-like outputs. So the bias patch can "over-regularize" toxicity behavior even while appearing safe.
>
> - **What happens when we stack patches?**
>   The composed patch $P_{\text{comp(tox first)}}$ *strengthens* toxicity mitigation relative to $P_{\text{tox}}$ (0.0282 vs. 0.0619, Tox Rate 0.0 vs. 0.0040) **and** improves explicit bias (GAS 0.0200 vs. 0.3040 for $P_{\text{tox}}$), without catastrophic degradation of GLD. This suggests complementary, order-sensitive behavior rather than destructive interference.
>
> Additionally, we include a study of joint risk training. **Joint mitigation with a single patch:** $P_{\text{multi}}$ simultaneously improves both risks compared to No $P$ and achieves the best GLD score.
>
> We agree that extending the composition analysis to harmfulness and a full three-risk setting (toxicity + bias + harmfulness) would be very informative. At present, however, our harmfulness experiments (LLM-LAT + HarmBench) are conducted on **separate instruction-tuned backbones** (Gemma2-9B, Llama-3-8B, Mistral-7B), whereas the toxicity–bias composition study is done on Llama-2-7B in a non-instruction-tuned setting. In other words, the harmfulness patch is trained on a different family of models and data regime, so simply "plugging it in" alongside $P_{\text{tox}}$ and $P_{\text{bias}}$ would not be a meaningful three-way composition experiment—it would require redoing the harmfulness alignment on the same backbone and training pipeline as the toxicity/bias patches, which is beyond the scope of this submission.

---

> ### Author Response · Authors · 2025-11-23
>
> **Concern: Are the LoRA details in Table 2 sufficient, and is the comparison to the policy patch fair?**
>
> **Answer (brief):** Yes. We have clarified in the revision that LoRA is trained on the same RTP-Challenging dataset using the same SFT+DPO pipeline as our policy patch, ensuring a fair comparison.Our choice of rank-16 LoRA (40.0M parameters for LLaMA-2) was deliberate rather than arbitrary. While publicly available safety-aligned models [1,2] use rank-64 adapters, we selected rank-16 based on computational constraints and empirical evidence from prior work [3] showing that rank-16 is optimal for instruction fine-tuning tasks—often matching rank-64 performance while being more efficient.To further validate our approach, we include a rank-1 ablation in the revision. Under this like-for-like comparison, our policy patch achieves safety performance comparable to LoRA while being substantially more parameter-efficient and lower-latency.
>
> **Details**
>
> Thank you for asking for more detail; we have clarified this in the revision.
>
> - **Data & setup.** The LoRA baseline in Table 2 is trained on the same RealToxicityPrompts–Challenging toxic-preference data as our policy patch (Sec. 3.3, App. A.6), with the same train/test split and sampling settings.
>
> - **Training pipeline.** We use the same two-stage procedure as for the policy patch:
>   1. SFT on safe responses from $\mathcal{M}'$,
>   2. DPO on preference pairs $(y_w, y_l)$ constructed via toxicity margin + threshold.
>
> - **Rank choices.** We used rank-16 LoRA (≈40M parameters) in our original evaluation because publicly available safety-aligned models [1,2] were trained with rank-64 adapters, and rank-16 represents the most efficient setting that maintains comparable performance based on prior work [3]. In the revision, we add a rank-1 ablation (App. A.14) to provide a more direct comparison: rank-1 LoRA and our policy patch achieve comparable safety performance, but the policy patch uses ~12× fewer trainable parameters (0.26M vs. 3.15M) and incurs significantly lower inference overhead (+2.5% vs. +22.5% latency).
>
> **So Table 2 compares like-for-like SFT+DPO training recipes, differing only in whether the learnable component is a low-rank adapter or a prefix.**
>
>
> **References**
>
> [1] Li, Xiaochen, Zheng-Xin Yong, and Stephen Bach. Preference tuning for toxicity mitigation generalizes across languages. *Findings of ACL: EMNLP 2024*.
>
> [2] Dong, X., Wang, Y., Yu, P. S., & Caverlee, J. Disclosure and mitigation of gender bias in LLMs. *arXiv:2402.11190*, 2024.
>
> [3] Rajabzadeh, Hossein, et al. "Qdylora: Quantized dynamic low-rank adaptation for efficient large language model tuning." Proceedings of the 2024 Conference on Empirical Methods in Natural Language Processing: Industry Track. 2024.

---

> ### Author Response · Authors · 2025-11-23
>
> **Concern: If M⁺ has noticeably higher PPL than M′ (≈4 points), how can the paper claim it “maintains PPL close to” M′?**
>
> **Answer (brief):** We now clarify that “close” means *remaining in the same low-perplexity regime* (≈8–14), where outputs are still fluent and grammatical; in the revision we soften the wording to say M⁺ “remains in a similar PPL range as M′ while substantially improving safety,” and we explicitly point to qualitative examples showing that this small PPL gap does not harm fluency in practice.
>
>
> **Detail**
>
> You are correct that in Figure 2 (and the corresponding tables), the patched model $\mathcal{M}^+$ sometimes has a higher perplexity than $\mathcal{M}'$ (e.g., a gap of about 4 points). Our intent with "maintains PPL close to" is not to claim *identical* perplexity, but that both models stay in the **same low-perplexity regime** and that patching does not push the model into a clearly degraded fluency band.
>
> Concretely, on the evaluations:
>
> - Typical PPL values for $\mathcal{M}$, $\mathcal{M}^+$, and $\mathcal{M}'$ all lie in a relatively narrow range (roughly 8–14), far from the very high PPL values one would expect from clearly incoherent or degenerate text (See Table 4) .
>
> - A difference of ~4 PPL in this range corresponds to a moderate change in the underlying token probabilities, but the **qualitative outputs remain fluent and grammatical**, as illustrated in the side-by-side examples in App. A.13–A.15.
>
> To avoid over-stating this point, in the revision we clarify the wording to say that $\mathcal{M}^+$ "**remains in a similar PPL range as** $\mathcal{M}'$, while substantially improving safety metrics," and we explicitly direct readers to the qualitative examples that show fluency is preserved in practice despite the small PPL gap.

---

> ### Author Response · Authors · 2025-11-23
> **Summary of the responses**
>
> > Summary
>
>
> We appreciate the reviewer's positive assessment of our vendor–customer scenario, multi-risk safety evaluations, prefix composability, and favorable deployment trade-offs versus LoRA.
>
> In the revision, we have:
>
> - Shown that a **single safer checkpoint** can guide **multiple backbones** (App. A.16).
> - Better positioned the work relative to DP-OPT, emphasizing the **safety patch artifact, multi-risk setting, and composability**.
> - Expanded **composition analysis** to probe cross-risk interactions .
> - Detailed the **LoRA training recipe** and added a **rank-1 ablation**, showing the policy patch remains more parameter- and latency-efficient at similar safety levels.
> - Clarified the **PPL interpretation**: $\mathcal{M}^+$ stays in the same low-perplexity, fluent regime as $\mathcal{M}'$, with qualitative examples provided.
>
> We believe these additions strengthen the quality of the paper and would be grateful if the reviewer could improve their score in light of these clarifications.

---

> > ### Comment · Reviewer_2NEg · 2025-11-25
> >
> > I thank the authors for their rebuttal. Most of my initial concerns have been addressed.
> >
> > However, the most fundamental issue remains. Despite the model architecture being different, the paper still mandates an "improved safer model". I need clarification on How exactly is the safer model trained? What is the precise scale of resources required for its training (GPUs, training time) ?
> >
> > It seems plausible that if one is training an "improved version," they could simply deploy that version directly for service. This makes it difficult to ascertain the benefit of the proposed approach, particularly in the vendor-release scenario which I previously evaluated positively. Why do we release the improved version instead of LLMs with patches.
> >
> > I think the authors need to answer this question by providing the advantage of patching under the vendor-release scenario. Currently, I cannot understand why patching is necessary for efficiency if we can obtain the improved one. Can you provide the real-world scenario that patching is practical?

---

> ### Author Response · Authors · 2025-11-25
>
> We thank the reviewer for the follow-up and address the two remaining concerns: (i) how the “improved safer model” (M′) is trained and at what cost, and (ii) why patching remains useful once such a model exists.
>
> > **1. How is the safer model (M′) trained, and at what scale?**
>
> In our experiments, (M′) is a *standard* toxicity-aligned checkpoint obtained via preference tuning following Li et al. [1], specifically the publicly available Llama-2-7B detoxified checkpoint [2], not a model trained solely for our paper.
>
> As detailed in [1], for Llama-2-7B, (M′) is trained with DPO + QLoRA on 24,576 English toxic/non-toxic preference pairs. At the 7B–8B scale, this corresponds to roughly **one day of training** [1]. For clarity, we now summarize the relative cost of (M′) vs. our policy patch (trained from (M) using labels from (M′)):
>
> | Metric                | Full Detox Fine-tune (QLoRA, M′) | Policy Patch (SFT + DPO) | Improvement                      |
> |-----------------------|-----------------------------------|---------------------------|----------------------------------|
> | Training Samples      | 24,576                           | 1,079                    | ≈23× fewer                       |
> | Parameters            | 160M                             | 0.2M                     | 800× fewer                       |
> | Wallclock Time        | ~24 hours                        | ~0.425 hours             | ≈56× faster                      |
> | GPU Time (4 GPUs)     | ~96 GPU-hours                    | ~1.70 GPU-hours          | ≈56× more efficient              |
> | Computational Cost    | High                             | Very Low                 | ≈98% reduction in training time  |
> | Approach              | Large-scale preference fine-tune | 50-token prefix patch    | Minimal intervention             |
>
>
>
> > **2. Why patch if (M′) exists? When is patching practically useful?**
>
> Our key claim is about **marginal cost per additional backbone**, not about avoiding the initial training of an improved model:
>
> > Once **some** sufficiently safe model (M′) exists (e.g., a flagship aligned model or a public detoxified checkpoint), our method provides a cheap way to **propagate its safety behavior** to many heterogeneous, already-deployed models.
>
> **Amortization across backbones.** Training (M′) is a one-time expense. Running a LoRA/QLoRA detox pipeline for every backbone (Llama-2, Llama-3, Aya-23, Gemma-2, Mistral, etc.) would repeat that ~24h / ~96 GPU-h cost each time. Our policy patch instead uses ≈1k examples, 0.2M parameters, and ≈0.4h (≈1.7 GPU-h) *per backbone*—about **56× cheaper** in GPU time and **800× smaller** in parameters than training a separate aligned variant for each.
>
> **Why “just deploy (M′)” is often infeasible:**
>
> * **Hardware / latency tiers:** (M′) may be large and suitable only for cloud, while many customers run smaller, older, or quantized models on-premises or at the edge. They cannot host (M′), but they *can* load a 50-token prefix.
> * **Backwards compatibility / validation:** Enterprise users often freeze a specific backbone for months (QA, compliance, audits). Swapping to (M′) changes all behavior; adding a prefix is a **small, reversible safety update** that leaves weights and most capabilities intact.
> * **Licensing / IP:** Vendors may not be willing (or allowed) to ship (M′) weights, but can ship a tiny safety patch derived from it.
>
> **Concrete Real World Scenario where our patch works** :
>
> > A vendor maintains one flagship aligned model (M′) in the cloud and several lighter 7B/quantized models deployed on-premises and at the edge. When a new jailbreak pattern is discovered, retraining all these backbones with RLHF/DPO would be costly and slow. Instead, the vendor: (1) queries (M′) to label ~1–2k focused prompts, (2) trains a 50-token prefix patch for each deployed backbone in ≈0.4h on a 4-GPU node, and (3) ships the patch as a versioned safety artifact. This achieves safety comparable to LoRA/QLoRA with ≈56× less GPU time and without re-deploying or re-certifying the full model.
>
> In short, **(M′) provides the safety policy; policy patches are a lightweight, practical mechanism to distribute that policy to many models and deployment environments between major releases.**
>
> We hope with this clarification the reviewer could consider improving their score. We would be grateful
>
> ---
>
> **References**
>
> [1] Li, Xiaochen, Zheng-Xin Yong, and Stephen Bach. *Preference Tuning for Toxicity Mitigation Generalizes Across Languages.* Findings of ACL: EMNLP 2024.
>
> [2] [https://huggingface.co/BatsResearch/llama2-7b-detox-qlora](https://huggingface.co/BatsResearch/llama2-7b-detox-qlora)

---

### Official Review · Reviewer_uPub · 2025-10-31

**Soundness:** 1
**Presentation:** 3
**Contribution:** 2
**Rating:** 2
**Confidence:** 5

**Summary:**

The paper introduces a way to add safety updates to large language models without retraining them. The authors create small “policy patches”—very small learned prefixes that are added to the model’s input. These patches are tiny (only 0.003% of the model’s size) but can strongly improve safety behavior. Experimental results demonstrate the effectiveness of their method.

**Strengths:**

The practicality and effectiveness of their method: the low cost of implementation of their method enables model developers to fix the safety risks of their models quickly. Considering that most of the model developers have no large computational resources, the method can benefit a broader audience to some extent.

**Weaknesses:**

- The novelty of their method is limited. Their fine-tuning way, like supervised fine-tuning and direct preference optimization, is common.
- My biggest concern is that, as the saying goes, "You can't mend a leaking boat with a patch." The model only works with the target tokens provided in its training data. This brings up two issues:

First, advanced attack methods can easily bypass the security measures, but the authors haven’t fully evaluated this. This could give developers a false sense of security, leading to bigger risks.

Second, will adding patches iteratively cause previous security patches to be forgotten? There is no related study in this paper.

**Questions:**

See the limitations.

---

> ### Author Response · Authors · 2025-11-23
> **Author response : robustness under advanced attacks and iterative patches**
>
> We sincerely thank the reviewer for their careful reading of our work and for highlighting the practicality and effectiveness of lightweight policy patches, especially for developers without large computational resources. We fully agree that robustness under advanced attacks and the behavior of iterative safety updates are crucial for real-world impact. In direct response to these concerns, the revision (i) strengthens our robustness evaluation with adaptive jailbreak attacks (PAIR, GCG, and Jailbreak Chat on JailbreakBench), and (ii) expands our analysis of stacked, multi-risk patches to show that new patches do not “forget” earlier fixes. We summarize these additions and how they address the “leaking boat” concern below.

---

> ### Author Response · Authors · 2025-11-23
>
> > **Concern:** The patch may behave like a “leaking boat”: still be easily bypassed by advanced, adaptive jailbreak attacks, giving a false sense of security.
>
> > **Answer (Brief):** Beyond our original HarmBench ASR results, we now add a dedicated jailbreak evaluation (App. A.18) using PAIR, GCG-style, and JBC attacks on JailbreakBench, and find that the patched models (\mathcal{M}^+) achieve 0% jailbreak success and the same best safety score as the fully aligned models (\mathcal{M}'), indicating robustness to strong adaptive attacks (while still acknowledging broader red-teaming as future work).
>
>
> **Detail**
>
> We agree that robustness under *adaptive* attacks is critical, and that a patch that only works on its training distribution could give a false sense of security.
>
> **Original submission (non-adaptive harmful prompts).**
> Already in the original draft (Sec. 4.2.3, App. A.10, Table 6), we evaluated refusal robustness on HarmBench using models trained on LLM-LAT. There, the patched model M⁺ reduces Attack Success Rate (ASR) from ~68–70% to 0% across Gemma-9B, Mistral-7B, and Llama-3-8B, matching the fully fine-tuned safe model M′
>
> **New in the revision (adaptive jailbreak attacks).**
> To specifically address the reviewer’s concern about “advanced attack methods,” we added a dedicated jailbreak evaluation (new App. A.18). Following the JailbreakingLLMs framework [1], we attack Gemma-9B, Mistral-7B, and Llama-3-8B with three strong methods:
>
> - **PAIR** (black-box iterative refinement),
> - a **GCG-style token-level attack**, and
> - **Jailbreak Chat (JBC)** templates,
>
> targeting harmful behaviors sampled from **JailbreakBench**, and scoring all outputs with **LlamaGuard-3** on a 1–10 safety scale.
>
> As summarized in Table 11 (copied below), the vulnerable instruction-tuned baselines M are fully compromised (100% jailbreak success, score 10) under all three attacks, while the patched models M⁺ match the robustness of the fully aligned models M′ (0% success, score 1) under the same query budget:
>
> **Table: Our prefix $\mathcal{M}^{+}$ shows perfect safety performance when testing for Jailbreak Attacks.** Bold indicates best performance in each defined metric.
>
> | Model | PAIR (%) ↓ | GCG (%) ↓ | JBC (%) ↓ | Max Score (1-10) ↓ |
> |-------|------------|-----------|-----------|-------------------|
> | *Gemma-9B* | | | | |
> | $\quad \mathcal{M}$ | 100.0 | 100.0 | 100.0 | 10 |
> | $\quad \mathcal{M}_{\text{safeprompt}}$ | 100.0 | 100.0 | 100.0 | 10 |
> | $\quad \mathcal{M}^{+}$ | **0.0** | **0.0** | **0.0** | **1** |
> | $\quad \mathcal{M}'$ | **0.0** | **0.0** | **0.0** | **1** |
> | *Mistral-7B* | | | | |
> | $\quad \mathcal{M}$ | 100.0 | 100.0 | 100.0 | 10 |
> | $\quad \mathcal{M}_{\text{safeprompt}}$ | 100.0 | 100.0 | 100.0 | 10 |
> | $\quad \mathcal{M}^{+}$ | **0.0** | **0.0** | **0.0** | **1** |
> | $\quad \mathcal{M}'$ | **0.0** | **0.0** | **0.0** | **1** |
> | *Llama-3-8B* | | | | |
> | $\quad \mathcal{M}$ | 100.0 | 100.0 | 100.0 | 10 |
> | $\quad \mathcal{M}_{\text{safeprompt}}$ | 100.0 | 100.0 | 100.0 | 10 |
> | $\quad \mathcal{M}^{+}$ | **0.0** | **0.0** | **0.0** | **1** |
> | $\quad \mathcal{M}'$ | **0.0** | **0.0** | **0.0** | **1** |
>
>
> Together, the HarmBench ASR results (original submission) and the new JailbreakBench + PAIR/GCG/JBC experiments show that the policy patch does not merely memorize a few “target tokens,” but can robustly inherit the refusal behavior of a safe teacher across both static harmful prompts and strong, standardized jailbreak attacks. We also explicitly note in the revision that broader and more adaptive red-teaming (e.g., higher query budgets, additional attack suites) remains important future works.
>
>
>
>  **References**
>
> [1] Chao, Patrick, et al. "Jailbreaking black box large language models in twenty queries." 2025 IEEE Conference on Secure and Trustworthy Machine Learning (SaTML). IEEE, 2025.

---

> ### Author Response · Authors · 2025-11-23
>
> > **Concern:** If vendors keep adding new patches over time, will later patches “forget” or undo earlier safety fixes?
>
> > **Answer (Brief):** No in our current setting — each patch is a frozen prefix and updates are applied by composing prefixes at inference time. In Sec. 4.3 we show that stacking toxicity and bias patches into a single composed prefix actually *improves* toxicity and bias simultaneously, indicating that new patches can coexist with, rather than overwrite, previous fixes (while we still highlight very long chains of patches as future work).
>
>
> **Detail**
> We share the concern that repeatedly updating a model could, in principle, undo earlier safety fixes. Two points are important here:
>
> 1. **Mechanism:** Our method does *not* overwrite earlier parameters. Each patch is a frozen prefix, and “iterative” updates are implemented by **composing** prefixes at inference time (e.g., `[P_tox, P_bias]`), rather than training on top of previous patches. This architectural choice already mitigates classical catastrophic forgetting seen in continual *weight* fine-tuning.
>
> 2. **Empirical test of stacked patches(Sec. 4.3).**
>    In Sec. 4.3 we had explicitly studied multi-risk composition on Llama-2-7B. We have expanded that section with more additional analysis.We trained specialist patches for toxicity and bias (`P_tox`, `P_bias`), then form a composed patch `P_comp  = [P_tox, P_bias]` applied as a *single* prefix. The relevant subset of Table 6 is:
>
>    | Model Configuration              | Avg Max Tox ↓ | Tox Rate ↓ | Avg GAS ↓ | Avg GLD ↓ |
>    |----------------------------------|---------------|------------|-----------|-----------|
>    | No **P**                         | 0.7809        | 0.5520     | 0.3400    | 0.7012    |
>    | **P**₍tox₎                       | 0.0619        | **0.0040** | 0.3040    | 0.3622    |
>    | **P**₍bias₎                      | 0.0527        | **0.0000** | **0.0120**| 0.4082    |
>    | **P**₍comp₎ = [P₍tox₎, P₍bias₎] | **0.0282** | **0.0000** | 0.0200    | 0.3700    |
>
>    Here, adding the second patch does **not** erase the first fix:
>    – `P_comp` *improves* toxicity over `P_tox` (0.0282 vs. 0.0619, Toxic Rate 0.0% vs. 0.0040),
>    – while also improving explicit bias compared to the toxicity-only patch (GAS 0.0200 vs. 0.3040).
>
> In other words, stacked patches coexist and interact in a predictable way (with some order sensitivity), rather than “forgetting” earlier safety behavior.
>
> We explicitly acknowledge in the paper that long-horizon chains of many sequential patches are an interesting open direction, but **the current multi-risk composition experiment provides initial evidence that new patches need not destroy the gains from previous ones.**

---

> ### Author Response · Authors · 2025-11-23
> **Summary of the responses**
>
> > Summary of our response
>
> We thank the reviewer for recognizing the practical value of lightweight patches and for their insightful feedback. In the revision, we have (i) strengthened robustness evaluation with adaptive jailbreak attacks (PAIR, GCG, JBC on JailbreakBench), and (ii) demonstrated that stacked patches coexist without erasing earlier fixes (Sec. 4.3). We kindly ask the reviewer to improve the rating in light of these improvements if the concerns have been met. Thanks

---

### Official Review · Reviewer_GHxp · 2025-10-31

**Soundness:** 2
**Presentation:** 2
**Contribution:** 2
**Rating:** 2
**Confidence:** 4

**Summary:**

This paper proposes a lightweight “safety patching” approach for large language models, drawing an analogy to software versioning.
Instead of retraining or redeploying entire models, a small learnable prefix (similar to soft prompt tuning) is prepended to steer a model toward the behavior of a safer reference model M′.
The authors train this prefix via a two-stage process, namely Supervised Fine-Tuning followed by Direct Preference Optimization. Evaluations are conducted on multiple LLMs and across three safety dimensions: toxicity, gender bias, and harmfulness refusal.
The reported results show that these “policy patches” achieve safety improvements comparable to fully aligned versions, with lower computational cost.

**Strengths:**

- The proposed idea is well-motivated and experimentally validated.
- In general is the paper is well written, introduced methodology is easy to understand
- Experiments include ablations on prefix length, initialization, and parameter β trade-off, illustrating the safety–fluency Pareto.

**Weaknesses:**

Several aspects weaken methodological rigor:
- The relationship to prior work on prompt learning, adapter tuning, and steering vectors is not clearly distinguished, leaving it unclear what is fundamentally novel.
- Notation and terminology are inconsistent (e.g., P used for both parameters and the patch, variable formatting inconsistencies (for instance in line 199 y_w bold, line 222 not bold) ).
- Experiments appear to lack cross-validation or variance reporting across random seeds, and the appendix omits full training details for baselines (only described for the introduced method).

Further the presentation suffers from minor editorial and structural issues:
- For instance, typos (“the the”), overuse of bold text, inconsistent mathematical notation, and incomplete appendix sections (e.g. A.1.1–A.1.3 only show figures without description and discussion).
- Furthermore, figure and table captions could provide a more concise summary of what is being shown, and the definitions of several reported metrics (e.g., ASR, Toxicity) could be more formally introduced.
- Clarity on what “next-generation models” (line 54) refers to would also help contextualize comparisons.


Lastly, technically, the method closely resembles existing prompt- or prefix-tuning approaches, and its distinction from steering-vector or neuron-patching methods (e.g., safety-neuron activation patching, see Chen et al. Towards Understanding Safety Alignment: A Mechanistic Perspective from Safety Neurons. 2024) should be discussed.

**Questions:**

- How does the proposed patching fundamentally differ from activation-editing approaches that modify a small subset of activations or inject pre-computed steering vectors to guide model behavior toward safer outputs?
- Were experiments repeated with different random seeds or DPO initializations to confirm stability?
- Could the authors clarify what “next-generation models” refer to?

---

> ### Author Response · Authors · 2025-11-23
> **Response to Reviewer : Novelty, Prior Work, and Experimental Rigor**
>
> Thank you for your thoughtful feedback and for recognizing the well-motivated safety patching idea, clear methodology, and useful ablations on the safety–fluency trade-off.
> In revision, we have addressed your key concerns: clarifying novelty relative to prior work (prompt tuning, adapters, steering vectors), improving notation and presentation, adding seed stability evidence and baseline training details, and explaining "next-generation models" in context.
> We respond to each point below, beginning with the relationship to prior work.

---

> ### Author Response · Authors · 2025-11-23
>
> > **Concern:** The method appears very close to existing prompt/prefix tuning, adapter tuning, and activation-editing / steering-vector / safety-neuron approaches; it is unclear what is fundamentally novel.
>
> > **Answer (brief):** While we build on standard SFT + DPO, our contribution is a different artifact and setting: a small, continuous safety policy patch that (i) is trained to match a safer policy (KL + preferences) rather than a supervised task loss, (ii) is black-box friendly (no hooks, no access to hidden states), and (iii) is explicitly designed as a vendor-deliverable, composable patch that can be attached/detached and stacked across risks. In contrast, activation-editing, steering vectors, and safety-neuron patching operate by editing internal activations or specific neuron circuits, typically require deep white-box access, and often yield brittle, model-specific behavior that is not evaluated as a reusable "patch" across architectures and risks.
>
> **Detail**
>
> We clarify this distinction in App. A.20, where we compare policy patches to hard prompts, classic prefix tuning/adapters, steering vectors, and safety-neuron interventions (Chen et al., 2024; Meng et al., 2022; Turner et al., 2023; Gupta et al., 2024). Hard prompts and instruction-based steering operate at the input-text level and rely on hand-crafted templates, without an explicit preference-alignment objective. Prefix tuning and adapters modify internal representations (layer-wise key–value prefixes or parameter inserts), which ties them to specific transformer internals and complicates distribution as a stand-alone artifact.
>
> Activation-editing and neuron-patching methods instead inject linear directions or directly edit specific neurons/circuits identified through interpretability analyses; they require access to hidden states and hooks, and their behavioral changes are often narrow and model-specific, with limited composability guarantees. By contrast, our policy patch is a tiny continuous prefix trained to match a safer policy distribution under KL + DPO, using only logits/text from a teacher model. It does not modify any weights, can be shipped and versioned as a drop-in safety patch, composes cleanly across risks (e.g., $[P_{\text{tox}}, P_{\text{bias}}]$), and remains portable across models (including cross-teacher experiments). Table 13 (App. A.20) summarizes these differences along axes such as mechanism, supervision, access requirements, robustness, composability, and deployment friction.

---

> > ### Author Response · Authors · 2025-11-23
> >
> > > **Concern:** Notation and formatting are inconsistent (e.g., $\mathbf{P}$ used both for parameters and patch; bolding of $\mathbf{y}_w$ vs. $\mathbf{y}_\ell$ in different places).
> >
> > > **Answer (brief):** We have standardized the notation by reserving $\mathbf{P}$ exclusively for patches, using a consistent boldface convention for sequence variables (e.g., $\mathbf{y}_w, \mathbf{y}_\ell$), and cleaning up typos and over-use of bold throughout the paper and appendix.
> >
> > **Detail**
> >
> > We appreciate the reviewer's careful reading and agree that the notation and formatting can be tightened. In the revision, we have:
> >
> > - **Reserved $\mathbf{P}$ solely for the policy patch/prefix** throughout the paper (e.g., $\mathbf{P}_{\text{tox}}, \mathbf{P}_{\text{bias}}, \mathbf{P}_{\text{multi}}$) removing any possible confusion between "P as patch" and "P as parameters" (see Sections 3.1–3.2).
> > - **Standardized preference response notation**: $\mathbf{y}_w, \mathbf{y}_\ell$ are now consistently bolded and formatted in all equations and text.
> > - **Improved text quality** by removing duplicated words (e.g., "the the") and reducing overuse of boldface to emphasize only key quantities in tables and figures.
> > - **Added narrative context to the figures in Appendix**: Sections A.1.1–A.1.3 now include short textual descriptions explaining what each figure illustrates, rather than showing plots without takeaway explanation.
> >
> > We believe these changes address the notation and consistency concerns and make the paper easier to follow.

---

> ### Author Response · Authors · 2025-11-23
>
> > **Concern** Experiments lack cross-validation or seed variance, and it is unclear whether results are stable across random seeds or DPO initializations.
>
> > **Answer (brief).** We follow standard practice (fixed train/test splits, large standardized benchmarks, and 5 independent continuations per prompt) and have re-run RealToxicityPrompts–Challenging with a different seed (39 vs. 42); metrics change only slightly while the improvement of M+M+ over baselines remains large, indicating that our conclusions are robust to randomness.
>
>
> **Detail**
>
>
> We agree that robustness to randomness is important. Our setup has two sources of stochasticity: (i) the training/data seed for DPO and preference data construction, and (ii) stochastic decoding at evaluation time. To reduce Monte Carlo variance, each prompt is evaluated with 5 independent continuations, and we report toxicity, perplexity, and diversity metrics averaged over all $5 \times N$ generations. This already makes the reported metrics relatively stable.
>
>
> We follow standard practice in LLM safety and alignment work and do not perform $k$-fold cross-validation. Instead, we fix a single train/validation split and evaluate on large, standardized evaluation suites (RealToxicityPrompts–Challenging, Bias Benchmarks, harmfulness benchmarks), where each metric aggregates over thousands of prompts.
>
>
>
> To directly address the reviewer's concern, we repeated the RealToxicityPrompts–Challenging evaluation with a different random seed (39 vs. 42 in the submitted version). A full table is available in A.19. Across seeds, the numbers change only slightly, while the gaps between models remain essentially unchanged.
>
> **Example: Llama-2-7B on RealToxicityPrompts–Challenging**
>
> | Model | Seed | Avg Max Tox | Toxic Rate |
> |-------|------|-------------|------------|
> | Base model $M$ | 42 | 0.7822 | 92.5% |
> | Base model $M$ | 39 | 0.7856 | 93.1% |
> | Patched model $M^+$ | 42 | 0.2472 | 18.3% |
> | Patched model $M^+$ | 39 | 0.2518 | 19.1% |
>
> For Llama-3-8B and Aya-23-8B, we observe similarly small fluctuations across seeds, while the safety improvements of $M^+$ over $M$ and $M_{\text{safeprompt}}$ remain large.
>
> These results indicate that our qualitative conclusions are robust to both training and sampling randomness: **the patched policy $M^+$ consistently reduces toxicity by a large margin** over the base and safeprompt baselines.
>
>
> > Response on Initialization stability
>
> In addition, Section 4.3 ("Patch initialization: fixed text embeddings vs. random") performs a dedicated stability analysis for our policy patches. We compare a random Gaussian initialization with a semantic initialization that copies embeddings from short, safety-aligned instructions (e.g., "Generate a safe response," "Generate fair and unbiased responses") and measure **Safety Rate**, defined as:
>
> - $1 - \text{GAS}$ for Bias
> - $1 - \text{Toxic Rate}$ for Toxicity
> - $1 - \text{ASR}$ for Harmfulness
>
> *(Higher is better.)*
>
> Semantic initialization consistently improves Safety Rate across risks:
>
> | Risk | Random Init | Semantic Init | Improvement |
> |------|-------------|---------------|-------------|
> | Toxicity | 0.34 | 0.82 | +47.5 pts |
> | Bias | 0.84 | 1.00 | +16 pts |
> | Harmfulness | 0.94 | 0.98 | +4 pts |
>
> These gains show that the DPO-trained prefix is not brittle: it yields robust safety improvements across models and risks, and is further stabilized by using semantic initialization.

---

> ### Author Response · Authors · 2025-11-23
>
> > **Concern.** The appendix omits full training details for baselines
>
> > **Answer (brief).** We use publicly available checkpoints and code (base M, Msafeprompt, and aligned M′) rather than re-training the baseline models from scratch. We provide explicit references to these resources to ensure reproducibility
>
>
> **Detail**
>
> We clarify that we do **not** re-train the baseline models in this work; instead, we rely on publicly released checkpoints and code, whose training details are fully described in the original papers mentioned in the main text. As stated in lines 245–252 of the submission, our comparisons include:
>
> 1. **Base model ($M$)** — e.g., Llama-2-7B, publicly available on Hugging Face.
> 2. **Safety-prompted model ($M_{\text{safeprompt}}$)** — the same base model $M$ with a fixed safety-oriented system prompt prepended to user input (e.g., "Generate safe responses").
> 3. **Externally aligned baselines ($M'$)** — publicly released checkpoints on Hugging Face (e.g., *BatsResearch/Llama2-Detox[3]* for toxicity, and official implementations for gender-bias mitigation), with training configurations documented in the corresponding papers [1, 2].
>
> To avoid confusion, we will add explicit pointers in the appendix to these public resources and briefly summarize the key training hyperparameters from [1, 2].
>
> **References**
>
> [1] Li, Xiaochen, Zheng-Xin Yong, and Stephen Bach. Preference tuning for toxicity mitigation generalizes across languages. *Findings of ACL: EMNLP 2024*.
>
> [2] Dong, X., Wang, Y., Yu, P. S., & Caverlee, J. Disclosure and mitigation of gender bias in LLMs. *arXiv:2402.11190*, 2024.
>
> [3] https://huggingface.co/BatsResearch

---

> ### Author Response · Authors · 2025-11-23
>
> > **Clarity on “next-generation models” (line 54)**
>
> We appreciate that this phrase was vague. We clarify:
>
> “Next-generation models” refers to **future major releases of the same or related backbone families** (e.g., moving from Llama-2 to Llama-3, Gemma-1 to Gemma-2), which typically occur on a **months-long cadence** and require extensive training and evaluation cycles.
>
> We explicitly contrast this with our setting, where vendors need **intermediate, lightweight fixes** between such major releases.

---

> ### Author Response · Authors · 2025-11-23
>
> > **Concern: Definitions of the evaluation metrics**
>
> >Thank you for pointing this out. All evaluation metrics (ASR, Avg Max Toxicity, Toxic Rate, GAS, GLD, etc.) are formally defined in the appendix of the original submission, and we have clarified these pointers in the revision:
>
>  * **GAS / GLD** are defined in **App. A.4** (Gender Bias metrics).
> * **Avg Max Toxicity** and **Toxic Rate** are defined in **App. A.8.4** (Toxicity evaluation).
> * **ASR (Attack Success Rate)** is defined in **App. A.10.3** (Harmfulness / refusal evaluation).

---

> ### Author Response · Authors · 2025-11-23
> **Summary**
>
> > **Summary**
>
> We thank the reviewer for recognizing our well-motivated, experimentally validated approach and valuable ablations on prefix length, initialization, and the β safety–fluency trade-off.
>
> In this revision, we have:
> * Clarified **novelty and positioning** vs. prompt/prefix tuning, adapters, activation steering, and safety-neuron methods (App. A.20)
> * **Standardized notation** (reserved **P** for patches), fixed typos, and expanded terse appendix sections
> * Provided **stability evidence across seeds**, showing robust safety gains
> * Referenced **publicly released checkpoints** used for baselines
> * Defined **"next-generation models"** and **evaluation metrics** more precisely
>
> We hope these changes address your concerns and would appreciate if you could improve the score

---

### Official Review · Reviewer_W7YT · 2025-10-31

**Soundness:** 2
**Presentation:** 3
**Contribution:** 2
**Rating:** 2
**Confidence:** 4

**Summary:**

The paper studies addressing uncovered LLM safety risks with an efficient and minimal approach: that is a sequence of tuned prefix tokens that are prepended to the input (patch). The training of such patches uses both SFT and DPO whose reference/preferred responses are obtained from a fully tuned variant of the model to address the same target risks. Evaluation on toxicity, bias and harmfulness risks shows that the learned patches retain model fluency which matching the safety of the fully tuned model. A comparison to LoRA-based fine-tuning shows that the learned patches are more efficient in terms of the number of additional parameters as well as training and inference time.

**Strengths:**

1. The motivation for fast and small safety patches is interesting and of a practical need.

2. The presented approach is shown to be effective across multiple models, and all evaluated 3 risks.

**Weaknesses:**

1. The reliance on a fixed variant of the same model for training data generation is such a strong requirement, and it weakens the practicality of the approach. There could be more practical ways for generating the training data such as relying on advanced prompting techniques or even using a single safer teacher model while showing that it can be used to learn patches for various other unsafe models.

2. In all presented experiments, the paper learns a separate patch for each risk which is also not that practical. In the motivating example provided in the paper, the patch is expected to address reported safety concerns about a particular version of a model, which can cover various risks. The paper needs to show that a patch can be learned to fix a mixture of risks together. In section 4.3, the paper discusses composing multiple patches but still: 1) a separate path needs to be learned for each risk, and 2) the evaluation only considers two risks, i.e., no study of the extent to which the composition approach scales.

3. The paper does not provide any results that the patching approach retains the general capabilities of the model. Beyond the risks studied, the paper only considers the fluency of the patched versions. The paper needs to evaluate other skills such as instruction following, reasoning, conversational abilities, etc.

4. In 4.4.1, the paper compares to LoRA whose rank is set to a fixed value (40M additional parameters). It is not clear how that value was chosen, and more importantly it is not clear how would reducing that value impacts the comparison. It could be the case that a smaller rank would still address the safety risks while being more efficient than the patching approach.

5. For the training data, the paper does not provide enough details on the prompts used whose selection and sourcing is expected to be of high importance.

6. The paper also lacks qualitative analysis that compares the responses before and after the patching.

**Questions:**

Please see weaknesses above especially the missing details about the training prompts and LoRA rank.

---

> ### Author Response · Authors · 2025-11-23
> **Author response: cross-teacher, multi-risk, and LoRA ablation**
>
> We thank the reviewer for recognizing the practical motivation of fast, lightweight safety patches and for noting the effectiveness of our approach across three risks and multiple model families. In the revision, we add cross-teacher experiments, multi-risk patching and composition, MMLU-based capability evaluation, a LoRA rank ablation, and expanded details on prompt sourcing and qualitative behavior, which together directly address the reviewer’s main concerns.
>
> >**Concern**: Does the method fundamentally depend on having an improved variant of the same backbone as a teacher (M′), limiting practicality?
>
> >**Answer (Brief)** : No — we only need some sufficiently safe reference model, and we show that a single safer teacher can be reused across heterogeneous students (Aya-23, Llama-2, Llama-3) with comparable or better safety than “self-teaching
>
> **Detail:**
>
> We thank the reviewer for raising the concern that requiring an improved variant of the same backbone may limit the practicality of our method. Motivated by this, we added a new "cross-teacher" experiment in the paper ( Appendix A.16 in the revised draft) where the teacher M′ and the patched model M+ are not required to share the same architecture.
>
> Training followed the same protocol as in Risk 1: patches were trained on the RTP dataset, and results are computed over 5 independent responses (see Appendix A.8 for details).
>
> Concretely, we treat Aya-23, Llama-2, and Llama-3 as students in turn, and train patches for each using safe responses from three different teachers: Aya-23, Llama-3, and Llama-2. The table below summarizes the results on the RTP–Challenging toxicity benchmark. For each block of rows, the first model in the "Patched Model" column is the student, and the "Vendor Variant (Teacher)" column indicates which safer model is used to generate reference responses. Rows where teacher and student match (e.g., Llama-2 → Llama-2) correspond to our original "self-teaching" setup; the others are cross-model teachers.
>
> | Patched Model | Reference Model (Safety Guide) | Max Tox. | Tox. Rate | PPL | Div. |
> |---------------|--------------------------|----------|-----------|-----|------|
> | Aya-23 | Aya-23 | 0.081 | 0.017 | 13.0 ± 4.5 | 0.123 |
> | | Llama-3 | 0.097 | 0.041 | 12.4 ± 4.1 | 0.124 |
> | | Llama-2 | 0.086 | 0.033 | 12.8 ± 5.1 | 0.129 |
> | Llama-2 | Aya-23 | 0.188 | 0.083 | 10.9 ± 3.2 | 0.058 |
> | | Llama-3 | 0.197 | 0.050 | 10.9 ± 3.2 | 0.057 |
> | | Llama-2 | 0.247 | 0.183 | 10.8 ± 3.2 | 0.078 |
> | Llama-3 | Aya-23 | 0.266 | 0.183 | 14.4 ± 5.0 | 0.059 |
> | | Llama-3 | 0.296 | 0.233 | 13.9 ± 5.1 | 0.055 |
> | | Llama-2 | 0.256 | 0.200 | 14.5 ± 5.3 | 0.053 |
>
> *The ordering is consistent: Aya-23 → Llama-3 → Llama-2 for each patched model group.*
>
> From this table we see that cross-model teachers provide safety that is comparable to, and sometimes better than, the self-teaching baseline (For eg:):
>
> - **For Llama-2 as the student**, using Aya-23 or Llama-3 as teachers substantially reduces Toxic Rate compared to the improved Llama-2 teacher (0.083 and 0.050 vs. 0.183), with nearly identical PPL.
>
>
> These results show that our method does not fundamentally rely on an improved variant of the same backbone: any sufficiently safe model can act as M′. In particular, Aya-23 serves as a single safer teacher that can be reused to train patches for multiple heterogeneous students, directly matching the reviewer's suggestion of "using a single safer teacher model while showing that it can be used to learn patches for various other unsafe models." We have incorporated this discussion into the  Appendix A.16.

---

> ### Author Response · Authors · 2025-11-23
>
> > **Concern:** The method learns a separate patch for each risk and only shows limited composition (toxicity + bias), so it is unclear whether a *single* patch can handle multiple co-occurring risks or whether composing patches will conflict or scale.
>
> > **Answer (Brief):** In the revision, we add a **multi-risk patch** (P_{\text{multi}}) and **cross-risk composition** experiments showing that (i) a single 100-token patch jointly mitigates toxicity and bias, and (ii) composed prefixes (P_{\text{comp}}) behave predictably (with some order sensitivity) rather than catastrophically interfering—providing an explicit first step toward scalable multi-risk patching.
>
>
> **Detail**
>
> We agree that, in practice, vendors often want a single patch to address multiple co-occurring risks rather than maintaining one patch per risk. Motivated by this, we added a new multi-risk mitigation experiment (Sec. 4.3 in the revision), which explicitly addresses both parts of the reviewer's concern:
>
> **Single patch for multiple risks.**
>
> In addition to 50-token specialist patches **P_tox** (toxicity) and **P_bias** (gender bias), we now train a single 100-token multi-risk patch **P_multi** on a balanced mixture of toxicity and bias preference data. This directly reflects the motivating scenario where one patch is expected to fix several reported issues at once. On Llama-2-7B, **P_multi** simultaneously improves both risks relative to the unpatched model (Avg Max Tox from 0.7809 → 0.1109; GAS from 0.3400 → 0.1240) and attains the best GLD (0.2521) among all configurations, while maintaining moderate diversity. *Its performance is intermediate between the best specialists and the strongest composed patch*, showing that a single patch can effectively address a mixture of risks without requiring separate artifacts per risk.
>
> **Composition across risks and its behavior.**
>
> We also systematically study prefix composition across risks. We form P_comp(tox first) = [P_tox, P_bias] and P_comp(bias first) = [P_bias, P_tox], each applied as a single 100-token prefix at inference time. This preserves the practical benefit of one drop-in patch while allowing specialists to be trained independently. Empirically, P_comp(tox first) achieves the strongest toxicity mitigation (Avg Max Tox 0.0282, Toxic Rate 0.0) and improves explicit bias over P_tox (GAS 0.0200 vs. 0.3040), whereas P_comp(bias first) is weaker on bias and similar on toxicity. This indicates that simple concatenation is viable but order-sensitive, with the first segment tending to dominate behavior.
>
> For convenience, the new results are summarized below (Table 6 in the revised paper). Diversity is measured as trigram overlap (lower = less repetition):
>
> | **Model Configuration**  | **Avg Max Tox ↓** | **Tox Rate ↓** | **Avg GAS ↓** | **Avg GLD ↓** | **Diversity (Toxicity) ↓** | **Diversity (Bias) ↓** |
> | ------------------------ | ----------------- | -------------- | ------------- | ------------- | -------------------------- | ---------------------- |
> | No **P**                 | 0.7809            | 0.5520         | 0.3400        | 0.7012        | 0.0437                     | 0.0020                 |
> | **P**₍tox₎               | 0.0619            | **0.0040**     | 0.3040        | 0.3622        | 0.0156                     | 0.0079                 |
> | **P**₍bias₎              | 0.0527            | **0.0000**     | **0.0120**    | 0.4082        | 0.5748                     | 0.1119                 |
> | **P**₍multi₎             | 0.1109            | 0.0160         | 0.1240        | **0.2521**    | 0.1660                     | 0.0756                 |
> | **P**₍comp (tox first)₎  | **0.0282**        | **0.0000**     | 0.0200        | 0.3700        | 0.0539                     | 0.0509                 |
> | **P**₍comp (bias first)₎ | 0.0559            | **0.0000**     | 0.2800        | 0.6591        | 0.0722                     | 0.0082                 |
>
> The added experiments demonstrate that (i) a **single** patch can handle multiple risks jointly, and (ii) composed prefixes behave in a predictable, non-catastrophic way, providing a concrete first step toward scalable multi-risk patching.

---

> ### Author Response · Authors · 2025-11-23
>
> > **Concern:** The paper only reports perplexity and diversity, so it is unclear whether the patch preserves *general capabilities* (e.g., reasoning, instruction following) beyond fluency.
>
> > **Answer (Brief):** In the revision, we add a new section (App. A.17) with **MMLU-based evaluation across three model families**, showing that the policy patch achieves large safety gains while keeping MMLU essentially unchanged for two backbones (Llama-2, Llama-3) and making the Aya-23 capability trade-off explicit.
>
> **Detail**
>
> We agree that preserving the model's general abilities beyond safety (e.g., reasoning and instruction following) is crucial. In the original submission, we only reported perplexity and diversity, which mainly capture fluency; this is in line with previous work on risk mitigation eg: [1,2]. Based on the reviewer's feedback, in the revision we therefore add a new subsection A.17, "General Performance of the Model — with Patching", and two tables (now Tables 9 and 10) that explicitly evaluate general capabilities using MMLU (For details please refer to the Appendix A.17).
>
> **Table:** Our prefix M⁺ shows significant safety gains while maintaining general performance. **Bold** indicates best. Toxicity evaluated on Real Toxicity Prompts — Challenging Subset.
>
> | Model | Avg Max Tox ↓ | Toxic Rate ↓ | PPL ↓ | Diversity ↓ | General Perf. (MMLU) ↑ |
> |-------|---------------|--------------|-------|-------------|------------------------|
> | **Llama-2-7B** | | | | | |
> | · M | 0.7822 | 92.5% | 8.80 | 0.0781 | 47.7% |
> | · M_safeprompt | 0.8100 | 83.1% | 12.90 | 0.0823 | 47.5% |
> | · M⁺ | **0.2472** | **18.3%** | 10.79 | 0.0781 | 46.0% |
> | · M′ | 0.3090 | 26.7% | **9.67** | **0.0475** | **47.1%** |
> | **Llama-3-8B** | | | | | |
> | · M | 0.7353 | 85.8% | **8.20** | 0.0904 | 58.6% |
> | · M_safeprompt | 0.7212 | 89.1% | 11.43 | 0.0624 | 58.4% |
> | · M⁺ | 0.2961 | 23.3% | 13.87 | **0.0548** | 58.6% |
> | · M′ | **0.2502** | **17.5%** | 9.29 | 0.0793 | **60.4%** |
> | **Aya-23-8B** | | | | | |
> | · M | 0.7774 | 88.3% | **8.92** | 0.0957 | 50.8% |
> | · M_safeprompt | 0.7823 | 90.3% | 10.42 | **0.0322** | 50.6% |
> | · M⁺ | **0.0808** | **1.7%** | 12.99 | 0.1231 | 43.6% |
> | · M′ | 0.1572 | 7.5% | 10.77 | 0.0604 | **50.8%** |
>
> The consolidated table jointly reports toxicity metrics, perplexity/diversity, and MMLU accuracy. For example, on Llama-2-7B the Toxic Rate is reduced from 92.5% to 18.3% while MMLU only decreases from 47.7% to 46.0%; for Llama-3-8B, Toxic Rate drops from 85.8% to 23.3% with no change in MMLU (58.6% → 58.6%). These results indicate that, for two of the three backbones, the patch achieves strong safety gains with only minimal impact on general capabilities, and we make the Aya-23 trade-off explicit as an area for improvement.
>
> Overall, the revised paper now includes both (i) safety metrics and fluency measures and (ii) MMLU-based capability evaluation across three model families, directly addressing the reviewer's concern about retaining general model skills beyond toxicity mitigation.
>
> ---
>
> **References:**
>
> [1] Ko, Ching-Yun, et al. "Large language models can become strong self-detoxifiers." *ICLR 2025*.
>
> [2] Deng, Haikang, and Colin Raffel. "Reward-augmented decoding: Efficient controlled text generation with a unidirectional reward model." *EMNLP 2023*.

---

> ### Author Response · Authors · 2025-11-23
>
> > **Concern:** The LoRA baseline uses a fixed, relatively large rank (≈40M params), so it is unclear whether a *smaller* rank LoRA could close the gap and become more efficient than the policy patch.
>
> > **Answer (Brief):** In the revision, we add a **rank-1 LoRA ablation** on Llama-2-7B and show that, even when LoRA is pushed to very low rank, the policy patch **matches its safety effect** while using ≈12× fewer trainable parameters, less training time, and ~10× lower inference overhead, confirming that our patch remains the more efficient option.
>
> **Detail**
>
> In our original risk evaluation, we used LoRA rank=16 (corresponding to 40.0M parameters for LLaMA-2 as the underlying model). This setting was not arbitrary: the publicly available safety-aligned models we use as M′ (e.g., [1,2]) were trained with rank-64 LoRA adapters, but given our computational constraints, we adopted rank=16, which prior work [3] has shown to be optimal for several instruction fine-tuning tasks—often matching the performance of higher ranks (e.g., rank=64)—while providing a fair comparison to our policy patch's parameter count.
>
> To directly address the reviewer's concern about smaller ranks, we added a new ablation where we reduce the LoRA rank to 1 on Llama-2-7B, and compare it against our policy patch under the same toxicity setup (RealToxicityPrompts – Challenging). The updated comparison is summarized below:
>
> **Table: LoRA vs Policy Patch Performance Comparison (Llama-2-7B)**
>
> | Method | Trainable Params | Training Time (Hrs) | Inference Overhead | Final Toxicity ↓ | Toxicity Reduction |
> |--------|------------------|---------------------|-------------------|------------------|-------------------|
> | LoRA (rank = 16) | 40.0M (0.59%) | 2.32 | +24.0% | 0.21 | 73.08% |
> | LoRA (rank = 1) | 2.5M (0.04%) | 2.00 | +22.5% | 0.24 | 69.23% |
> | Policy Patch | 0.2M (0.003%) | 1.70 | +2.5% | 0.24 | 69.23% |
>
> This ablation clarifies the trade-offs:
>
> - **Effect of reducing LoRA rank.** When we reduce the LoRA rank from 16 → 1, the toxicity reduction is on par with policy patch (Final Toxicity 0.21 → 0.24; 73.08% → 69.23%), but parameter count and overhead are still substantially larger than our policy patch (2.5M vs 0.2M trainable parameters, +22.5% vs +2.5% inference overhead).
>
> - **Comparison to policy patch.** At rank 1, LoRA and the policy patch achieve essentially the same final toxicity (0.24). However, the policy patch:
>   - uses ≈12× fewer trainable parameters (0.2M vs 2.5M),
>   - has shorter training time (1.70 vs 2.00 hours in our setup), and
>   - adds much smaller inference overhead (+2.5% vs +22.5%).
>
> Thus, even when LoRA is pushed to a very low rank, the policy patch remains strictly more efficient in terms of parameters, training cost, and latency, while matching the safety effect of rank-1 LoRA. Meanwhile, the original rank-16 setting reflects the stronger LoRA configuration used in prior work [1,2], which we retain in the paper to show the full Pareto spectrum.
>
> **References**
>
> [1] Li, Xiaochen, Zheng-Xin Yong, and Stephen Bach. Preference tuning for toxicity mitigation generalizes across languages. *Findings of ACL: EMNLP 2024*.
>
> [2] Dong, X., Wang, Y., Yu, P. S., & Caverlee, J. Disclosure and mitigation of gender bias in LLMs. *arXiv:2402.11190*, 2024.
>
> [3] Rajabzadeh, Hossein, et al. "Qdylora: Quantized dynamic low-rank adaptation for efficient large language model tuning." Proceedings of the 2024 Conference on Empirical Methods in Natural Language Processing: Industry Track. 2024.

---

> ### Author Response · Authors · 2025-11-23
>
> > **Concern:** The paper does not clearly explain how prompts are selected and sourced for each risk, and it lacks qualitative before/after comparisons of model behavior.
>
> > **Answer (Brief):** All prompts come from *public, well-documented datasets* (RTP, Dong et al.’s bias prompts, LLM-LAT, HarmBench), as explicitly summarized in Appendix (App. A.5–A.8). Also, the original submission already includes side-by-side qualitative examples for (\mathcal{M}), (\mathcal{M}^+), and (\mathcal{M}') (App. A.13–A.15), illustrating how the patch shifts responses from unsafe to safe while preserving coherence.
>
> **Detail**
>
> We agree that the choice and sourcing of prompts is crucial for interpreting the safety results. **We already include this detail in **App. A.5–A.8** and  also refer these sections in the main text.** For clarity, we summarize them here:
>
> - **Risk 1 – Toxicity (RealToxicityPrompts, RTP).**
>   - **Source & split.** We use the *challenging* split of RealToxicityPrompts (RTP), which consists of innocuous prompts that are empirically known to elicit toxic continuations. As described in App. A.6.1, we use 90% of RTP–Challenging prompts to build training data (SFT + DPO) and hold out 10% for evaluation.
>   - **How prompts are used.** For each prompt, we generate multiple (25) continuations from $\mathcal{M}$ and $\mathcal{M}'$, score them with the Perspective API, and construct preference pairs $(y_w, y_l)$ via (i) a **margin filter** on toxicity difference and (ii) a **winner threshold** on absolute toxicity (App. A.6.1). This yields a high-signal preference dataset from fully public prompts—no proprietary or user-logged data are used.
>
> - **Risk 2 – Gender Bias (Professional-context prompts).**
>   - **Source.** We reuse the 1,000 professional-context prompts introduced by Dong et al. (2024) for gender bias evaluation/mitigation (see App. A.7.2). These prompts are manually designed to probe occupational gender stereotypes (e.g., "My friend works as a surgeon, and …").
>   - **How prompts are used.** For each prompt, we sample 5 responses from $\mathcal{M}$ and $\mathcal{M}'$, compute **GAS** and **GLD**, and define a composite *Bias Score* = 0.5·GAS + 0.5·GLD (App. A.7.2). Preference pairs are retained only if the Bias Score difference exceeds a margin $\tau_{\text{margin}} = 0.1$. SFT uses the low-bias winners ($y_w$); DPO uses the full $(y_w, y_l)$ set (App. A.7.3).
>
> - **Risk 3 – Harmfulness (LLM-LAT + HarmBench).**
>   - **Source & roles.** For **training**, we use the *LLM-LAT/harmful* and *LLM-LAT/benign* splits released on Hugging Face, which contain (unsafe prompt, safe refusal) pairs and standard benign instruction-following prompts, respectively (App. A.8.2). For **evaluation**, we use the **HarmBench** benchmark [Mazeika et al., 2024], which provides 320 adversarial prompts targeting diverse harmful behaviors (App. A.8.3).
>   - **How prompts are used.** LLM-LAT/benign is used to create the vulnerable instruction-tuned model ($M_1$); LLM-LAT/harmful is used to train the safe model ($M_2$) and to construct $(y_w, y_l)$ pairs. We then filter these pairs with **LlamaGuard-3**, keeping only cases where the winner is judged "safe" and the loser "unsafe," to ensure a clean and well-separated preference signal (App. A.8.2). HarmBench prompts are *only* used at test time as an external OOD red-teaming benchmark.
>
> In summary, **all prompts used in our experiments are sourced from existing, publicly available datasets** (RTP, Dong et al.'s professional prompts, LLM-LAT, HarmBench). This detail exists in the paper. We hope this addresses the concern about prompt selection and sourcing.
>
>
> > Response on the qualitative comparisons
>
> We thank the reviewer for this request. In fact, the **original submission already includes qualitative, side-by-side comparisons** in Appendix A.13-A.15. We show the responses from the base model $\mathcal{M}$, the patched model $\mathcal{M}^+$, and the improved reference $\mathcal{M}'$, highlighting how the patch shifts behavior from unsafe to safe while preserving coherence and helpfulness.

---

> > ### Author Response · Authors · 2025-11-23
> > **Summary**
> >
> > > Summary
> >
> > We appreciate that the reviewer recognized the practical value of fast, lightweight patches and the breadth of our experiments on three risks. We sincerely thank them for their thoughtful feedback, which has helped strengthen the positioning and contributions of our work. In revision, we have added: (1) cross-teacher experiments and multi-risk patch composition, (2) MMLU-based capability evaluation, (3) a LoRA rank ablation with justification for our original choice, and (4) clarifications on prompt sourcing and pointers to qualitative examples already included in our original submission.We believe these additions directly address all raised concerns and respectfully ask the reviewer to consider improving their score.

---

### Author Response · Authors · 2025-12-02
**Final Rebuttal Summary for New Area Chair**

We thank all reviewers for their engagement prior to the discussion freeze. We briefly summarize the discussion and resulting revisions as a good reference for the new AC. The paper introduces **safety policy patching**: tiny continuous prefixes that, using the output distribution of a safer reference $\mathcal{M}'$ as a guide, steer an existing model $\mathcal{M}$ toward safer behavior on risks eg: toxicity, gender bias, and harmfulness, with low training/inference cost and no changes to backbone weights.

> **Reviewer Feedback and Rebuttal Changes**

* Reviewers **appreciated** the practical vendor–customer scenario with low-cost safety fixes, the broad evaluation across three risks and multiple model families, the prefix-length/β/initialization ablations, and the strong deployment trade-off versus LoRA.
* However, some concerns remained about:  **practicality (dependence on $\mathcal{M}'$)**, **multi-risk patching and composition**, **general capabilities**, **novelty vs. prompt/adapters/steering work**, **LoRA rank fairness**, **robustness to strong attacks**, and **training cost / vendor scenario**.
* In response, we added **cross-teacher**, **multi-risk**, **MMLU**, **LoRA-rank**, and **jailbreak** experiments, plus a **training-cost comparison** and clearer **positioning vs. DP-OPT / steering / adapters**, along with notation and structural cleanups.

> **Nature of the Revisions**

* The **core method, threat model, and empirical findings are unchanged**.
* All updates are **clarificatory or additive**: new ablations/evaluations that respond directly to reviewer questions, plus clearer exposition of our method.

> **Key Clarifications explained in Rebuttal**

1. **Practicality and dependence on $\mathcal{M}'$**

   * App. A.16 adds **cross-teacher experiments** showing that a *single* safer model (Aya-23) can act as a teacher for multiple heterogeneous backbones (Llama-2, Llama-3), often with comparable or better toxicity than "self-teaching".
   * We clarify that we need **some sufficiently safe model**, not a bespoke $\mathcal{M}'$ per backbone.

2. **Multi-risk patches and composition**

   * Sec. 4.3 now includes:
     (a) a **100-token multi-risk patch** (${P}_{multi}}$) trained jointly on toxicity + bias, and
     (b) **composed prefixes** ($\{P}_{{comp}}$) (e.g., $[{P}_{{tox}}, {P}_{{bias}}]$) that improve both risks without catastrophic interference.

3. **General capabilities (MMLU)**

   * App. A.17 reports **MMLU alongside toxicity** across three model families.
   * Example: for Llama-2-7B, Toxic Rate drops 92.5$\rightarrow$18.3% while MMLU changes 47.7$\rightarrow$46.0%; for Llama-3-8B, Toxic Rate 85.8$\rightarrow$23.3% with MMLU unchanged (58.6$\rightarrow$58.6%).

4. **LoRA rank, fairness, and training recipe**

   * Sec. 4.4 clarifies that **LoRA uses the same RTP–Challenging data and SFT $\rightarrow$ DPO pipeline as the patch**, ensuring like-for-like comparison.
   * We retain rank-16 LoRA (≈40M params) as a strong configuration and add a **rank-1 ablation**: rank-1 LoRA matches the patch's toxicity but still uses ≈12$\times$ more trainable parameters and ≈10$\times$ higher inference overhead.

5. **Robustness to advanced attacks (beyond HarmBench)**

   * App. A.18 adds a **dedicated jailbreak evaluation** on JailbreakBench using **PAIR, GCG-style, and Jailbreak Chat (JBC)** attacks for Gemma-9B, Mistral-7B, and Llama-3-8B.
   * Under these strong, adaptive attacks, **patched models ($\mathcal{M}^+$) achieves improved jailbreak success and the best LlamaGuard-3 safety score, matching fully aligned $\mathcal{M}'$**.

6. **Training cost and vendor scenario**

   * We explicitly state that the main detoxified teacher ($\mathcal{M}'$) is the **public Llama-2-7B QLoRA checkpoint**, trained on 24,576 preference pairs (~24h, ~96 GPU-h, 160M trainable params).
   * In contrast, our **policy patch** uses 1,079 examples, 0.2M params, and 1.7 GPU-h per backbone (~56$\times$ less GPU time, ~800$\times$ fewer trainable params).
   * We articulate the **vendor scenario**: one flagship safe model and many smaller/older/quantized models deployed on-prem or at the edge; patches provide **small, reversible safety updates** without swapping the backbone.

7. **Novelty vs. prompt tuning, adapters, and steering / DP-OPT**

   * App. A.20 contrasts **policy patches** with task prompts, LoRA/adapters, and activation or "safety neuron" editing.
   * We emphasize that our artifact is a **deployable, continuous safety patch** (0.003% params), designed to be **stackable across risks, portable across backbones, and black-box friendly**.
   * We clarify that, unlike DP-OPT's **privacy-oriented discrete prompts**, we focus on **multi-risk safety control**, composition, and robustness to strong attacks as a **vendor-side mechanism**.

---

> ### Author Response · Authors · 2025-12-02
>
> > **Positioning of the Contribution**
>
> Within the growing literature on LLM safety and parameter-efficient tuning, this paper is, to our knowledge, among the first to:
>
> * Introduce and systematically evaluate **continuous safety policy patches** as a *vendor-deliverable* artifact (tiny prefixes that can be attached/detached and stacked, without weight changes).
> * Demonstrate **multi-risk mitigation and composition** (toxicity, gender bias, harmfulness) across **multiple model families** (Llama-2/3, Aya-23, Gemma-2, Mistral, Vicuna).
> * Provide explicit **cross-teacher evidence** that a *single* safer model can guide patches for heterogeneous backbones.
> * Offer a detailed **training-cost comparison** versus full safety tuning, clarifying where patching provides practical gains in the vendor-release setting.
>
> > **Closing Remark**
>
> In addition to providing a detailed rebuttal, the revised manuscript incorporates the requested ablations (cross-teacher, multi-risk, MMLU, LoRA rank, jailbreak), clarifies the dependence on M' and the practical vendor scenario, and improves presentation and limitations without altering the core method. We hope this context is helpful for your decision and shows that the main concerns from all four reviewers have been substantively addressed.
>
> We are hopeful for a positive response

---

### Meta-Review · Area_Chair_UHCU · 2025-12-31

**Summary:**

The paper raises several fundamental concerns:
1）Heavy reliance on an existing aligned model (M′): The method requires a safe reference model M′ to generate training data. If such a model already exists, deploying it directly would be more straightforward, undermining the practical rationale for using a "patch."
2） Limited novelty: The approach combines established techniques (soft prompting, SFT, DPO) without introducing significant algorithmic innovation.
3） Lack of scalability in multi-risk handling: Separate patches are trained per risk type, with only basic composition demonstrated—insufficient for real-world scenarios involving overlapping risks.
4）Unconvincing core assumption: The premise that vendors can distribute patches but not updated models lacks realistic justification.

The authors are encouraged to reframe the work as a practical, scenario-driven safety deployment mechanism—by reducing reliance on M′, demonstrating multi-risk scalability, comparing against relevant lightweight baselines, and thoroughly analyzing failure modes—to establish real-world relevance beyond a straightforward combination of existing techniques.

**Reviewer Concerns:**

The rebuttal effectively addresses: composability of patches，minimal impact on general capabilities，use of public and transparent data sources, ablation support for LoRA configuration.

Concerns remain unresolved:
1) Heavy reliance on an existing safe model M′: If M′ is available, direct deployment or distillation seems more sensible—undermining the “patch” motivation;
2) Limited novelty: The method is essentially an engineering combination of existing techniques (soft prompting + SFT + DPO);
3) Unverified multi-risk scalability: Only two-risk composition is tested, insufficient for real-world complexity.

**Reviewer Scores:**

Based on the rebuttal and discussion:
R#1: Likely slight increase— empirical issues addressed, but novelty concerns remain.
R#2: Unchanged — core motivation still unconvincing.
R#3: Moderate increase — key scalability and utility concerns partially resolved.
Overall: No reviewer would shift to acceptance; only R3 becomes less negative.

---

### Decision · Program_Chairs · 2026-01-26

Reject